# An Oral FMT Capsule as Efficient as an Enema for Microbiota Reconstruction Following Disruption by Antibiotics, as Assessed in an In Vitro Human Gut Model

**DOI:** 10.3390/microorganisms9020358

**Published:** 2021-02-11

**Authors:** Cécile Verdier, Sylvain Denis, Cyrielle Gasc, Lilia Boucinha, Ophélie Uriot, Dominique Delmas, Joël Dore, Corentin Le Camus, Carole Schwintner, Stéphanie Blanquet-Diot

**Affiliations:** 1UMR 454 MEDIS, Université Clermont Auvergne, INRAe, F-63000 Clermont-Ferrand, France; cecile.verdier@uca.fr (C.V.); sylvain.denis@uca.fr (S.D.); ophelie.uriot@uca.fr (O.U.); 2MaaT Pharma, F-69007 Lyon, France; cgasc@maat-pharma.com (C.G.); lboucinha@maat-pharma.com (L.B.); dominique.delmas@valotec.com (D.D.); joel.dore@inrae.fr (J.D.); clecamus@maat-pharma.com (C.L.C.); cschwintner@maat-pharma.com (C.S.); 3MICALIS and MétaGénoPolis, Université Paris Saclay, INRAe, AgroParisTech, F-78350 Jouy-en-Josas, France

**Keywords:** gut microbiota, antibiotic, dysbiosis, FMT, enema, oral capsule, in vitro gut model

## Abstract

Fecal microbiota transplantation (FMT) is an innovative therapy already used in humans to treat *Clostridioides difficile* infections associated with massive use of antibiotics. Clinical studies are obviously the gold standard to evaluate FMT efficiency but remain limited by regulatory, ethics, and cost constraints. In the present study, an in vitro model of the human colon reproducing medically relevant perturbation of the colonic ecosystem by antibiotherapy was used to compare the efficiency of traditional FMT enema formulations and a new oral capsule in restoring gut microbiota composition and activity. Loss of microbial diversity, shift in bacterial populations, and sharp decrease in fermentation activities induced in vivo by antibiotherapy were efficiently reproduced in the in vitro model, while capturing inter-individual variability of gut microbiome. Oral capsule was as efficient as enema to decrease the number of disturbed days and bacterial load had no effect on enema performance. This study shows the relevance of human colon models as an alternative approach to in vivo assays during preclinical studies for evaluating FMT efficiency. The potential of this in vitro approach could be extended to FMT testing in the management of many digestive or extra-intestinal pathologies where gut microbial dysbiosis has been evidenced such as inflammatory bowel diseases, obesity or cancers.

## 1. Introduction

The digestive tract harbors the largest and most complex microbial community of the human body, namely the gut microbiota, which is mainly composed of thousands of bacterial species but also members of Archaea, Eukaryotes, and viruses [1,2,3]. The highest bacterial density is found in the large intestine with up to 10^12^ cells per gram of intestinal content [4]. The gut microbiota plays a fundamental role for the host under normal homeostasis, and is involved in physiological, nutritional, and immunological processes [4]. A large number of studies in animal models and humans have shown that a persistent imbalance in gut microbial community is associated with intestinal disorders such as inflammatory bowel diseases or irritable bowel syndrome or even extra-digestive pathologies like diabetes, obesity, cancers or neurological disorders. This alteration, named dysbiosis, has been associated with loss of richness and diversity, loss of keystone taxa, shift in metabolic pathways and/or bloom of pathobionts like *Enterobacteriaceae* or *Clostridiaceae* [4,5]. This raises a number of questions such as the causal relationship between disorder and gut microbiota alteration or if a return to an equilibrium state would be sufficient to eradicate or alleviate symptoms and/or how to come back to a “healthy” state.

Fecal microbiota transplantation (FMT) is the administration of fecal material from a healthy donor into the intestinal tract of a patient. FMT is performed in order to restore an altered microbiome, get back to a healthy state and thus confer a health benefit. FMT was reported to be used in traditional Chinese medicine 1700 years ago as a “yellow soup” to treat severe diarrhea [6,7] and its first description in English literature was recorded in 1958 to treat pseudomembranous colitis [8]. To date, FMT is a successful treatment option for patients with recurrent or refractory *Clostridioides difficile* infection (rCDI), a major cause of antibiotic-associated diarrhea [9,10]. However, FMT has also promising therapeutic value in several other disorders associated with gut microbial dysbiosis such as inflammatory bowel diseases, irritable bowel syndrome, cancers, acute myeloid leukemia, graft-versus-host disease, neurodegenerative disorders, autism, obesity and others [4,10,11,12]. FMT can be divided into two types: autologous FMT using patient’s own feces (stool collection being performed before deleterious treatment) and allogenic FMT with the use of related or unrelated healthy donor fecal samples [13,14]. While autologous FMT has proven its efficiency in restoring intestinal microbiota composition following disruption by antibiotics use during allogeneic hematopoietic stem cell transplantation [15], allogenic FMT was found to be most efficient in case of rCDI [16]. Several routes of FMT administration are available, namely enema, colonoscopy, naso-gastric duodenal, jejunal infusion or oral capsule [6,7,17]. Each approach presents some limits such as difficulties to retain administered suspension for enema, risk of vomiting and aspiration as well as discomfort during administration for naso-gastric tubes and required sedation and risk of tissue perforation for jejunal infusion and colonoscopy [18]. Oral capsules were recently developed to address gaps and limitations previously observed in other FMT routes of administration. Oral capsules offer the least invasive, cheapest and most easily stored administered form and eliminate several procedural risks encountered with the other routes of FMT treatment. However, those capsules have been associated with side effects—such as risk of vomiting and aspiration—and sometimes they can fail to reach their intestinal target [14,17]. In addition, Kao et al. have shown that oral capsules are non-inferior to delivery by colonoscopy and represent an effective approach in the treatment of rCDI [19].

Obviously, clinical studies remain as the gold standard approach to evaluate the effect of FMT treatment on gut microbial restoration and determine associated health benefits. Nevertheless, this approach may be hampered by heavy regulations, ethical concerns and high experimental cost. In order to reduce human experiments, animal models can be used in preclinical phases to evaluate FMT efficiency. Studies mainly involve conventional mice models but also Dextran Sodium Sulfate-induced colitis mice models or even human microbiota-associated rodent models [20,21,22,23]. These models have been advantageously used to assess FMT effect following antibiotic treatment or chemotherapy [20] but also as an alternative strategy in inflammatory bowel disease or in the management of metabolic disorders [21]. In vivo models present the great advantage to integrate host biological responses after FMT treatments such as weight gain/loss, inflammatory response, epithelial barrier integrity or host cell receptor activation. However, this in vivo approach in rodent remains limited by increased ethical and regulatory constraints and also by the high level of expertise required to handle animals (especially human microbiota-associated mice). Furthermore, significant differences in terms of both diet and digestive physiology, including gut microbiota, have been found between animals and humans [24]. FMT is also often administered by force-feeding in rodents which is far from the rectal administration frequently used in clinical practices [20,22].

A relevant alternative to animal experiments in preclinical phases is the use of in vitro models simulating the human digestive environment. Few studies have already used in vitro models of the human colon for investigating the effect of FMT on *Clostridioides difficile* [25,26]. Here, we describe for the first time the use of a human colonic system to assess the efficiency of different FMT formulations. For this purpose, we used the well-validated ARtificial COLon (ARCOL) system which integrates the main physicochemical and microbial parameters of the human colonic environment based on in vivo data, i.e., pH, temperature, retention time, supply of a nutritive medium reproducing ileal effluent composition, complex and metabolically active colonic microbiota and anaerobiosis maintained by the sole activity of resident microbiota [27,28]. Perturbations of gut microbiome were first induced in ARCOL by an antibiotic (ATB) treatment with ciprofloxacin, frequently used in humans to treat a wide range of bacterial infections. Then, the “dysbiotic” in vitro model was used to compare the efficiency of two enema dosages (10 g and 30 g) and a new autologous FMT capsule in restoring gut microbiota composition and metabolic activity (gas and short chain fatty acids—SCFAs—production).

## 2. Materials and Methods

### 2.1. Fecal Sampling and FMT Preparation

Fresh fecal samples were collected from three healthy adult human volunteers (a 27-year-old woman and two men aged 35 and 50 years) with no history of antibiotic or probiotic treatment 3 months prior to the beginning of the study. Fecal samples were kept anaerobically for a maximum of 6 h before treatment. Fecal inocula for the in vitro colon model were prepared using 55 g of fresh fecal samples under strict anaerobic conditions in a vinyl anaerobic chamber (Coy Laboratory Products Inc, Grass Lake, MI, USA). Stools were mixed with 500 mL of a 30 mM anaerobic sodium phosphate buffer (pH 6.5) supplemented with 1.9 mM cysteine, then the fecal suspension was filtered through a double layer of gauze. The fecal suspension was divided into 100 mL aliquots that were rapidly transferred into each of the five bioreactors simultaneously inoculated.

Filtrates of the same fecal suspensions were used by MaaT Pharma (Lyon, France) to prepare autologous FMT enema and capsules, according to its manufacturing operating system. FMT enema is a fecal-microbiota suspension for rectal administration stored in a special bag at −80 °C (volume of 128–150 mL for 30 g enema and 43–50 mL for 10 g enema). The 30 g and 10 g enemas contain about 30 × 10^11^ and 10 × 10^11^ bacteria, respectively, with a bacterial viability superior to 50%. FMT capsule (0.45 g) is a caecum-release capsule containing the freeze-dried form of the enema formulation. A capsule contains about 0.35 × 10^10^ bacteria, with a bacterial viability superior to 50%. For FMT capsules, only the active ingredient (enema in its freeze-dried form) was introduced into ARCOL model.

### 2.2. In Vitro Artificial Colon System ARCOL

Human colonic conditions were simulated in ARCOL model using MiniBio 500 mL my-Control bundles and Lucullus^®^ Lite software from Applikon (Delft, The Netherlands). Fermentations were conducted under semi-continuous conditions. The in vitro system reproduces, based on in vivo data, the main physicochemical and microbial conditions encountered in a healthy human adult colon [27,29,30]. Briefly, at the beginning of the experiment, fecal suspension (100 mL) was added into the bioreactor already filled with 200 mL of nutritive medium while flushing with O_2_-free N_2_ gas. Afterwards, anaerobic conditions were maintained exclusively through the sole activity of the resident microbiota and by ensuring the system airtightness. Overproduced gases were collected in a gas sampling bag connected to the condenser. Fermentation temperature was set at 37 °C and maintained inside the bioreactor using an incorporated panel heater. Colonic pH and redox potential were constantly recorded (Applikon, Delft, The Netherlands) and pH was adjusted to a value of 6.3 with an automatic addition of 2 M NaOH. The amount of NaOH consumed was recorded daily. After one day of batch fermentation, the nutritive medium containing various sources of carbohydrates, proteins, lipids, minerals, and vitamins (Table 1) to closely mimic the composition of human ileal effluents (nutrient availability and biliary salts concentration), was continuously introduced into the bioreactor at a flow rate of 0.21 mL/min. The fermentation medium was stirred at a constant speed of 400 rpm. Its volume was monitored using a level sensor and maintained at a constant value of 300 mL by automatic withdrawal of the fermentation medium, ensuring a mean retention time of 24 h. This set-up with colonic parameters allows a shift from fecal to colonic microbial profiles [31].

### 2.3. Experimental Design of In Vitro Fermentations

Five bioreactors run in parallel were inoculated with the fecal suspension from one donor (Figure 1) and used as follows: the first bioreactor was used as a control with no antibiotic treatment (control) while the second bioreactor was treated with ciprofloxacin (Sigma-Aldrich, Saint-Louis, MO, USA, 17850-5G-F), with an initial addition of 150 mg on day 6 followed by a continuous supply of 500 µg/mL in the nutritive medium up to day 12 (ATB control). The three other bioreactors were treated with ciprofloxacin as described for ATB control and then received: 27 mL of enema preparation at day 14 and day 15 (30 g enema), 9 mL of enema preparation at day 14 and day 15 (10 g enema) or the content of 3 capsules per day for 7 consecutive days from day 14 to day 20 (capsule). The total amount of bacteria administered in ARCOL model with FMT treatment is 1.2 × 10^12^, 0.4 × 10^12^, 1.47 × 10^10^, for 30 g enema, 10 g enema and capsule, respectively. Experiments were performed in triplicate (biological replicates named Run1, Run2 and Run3) with fecal samples from each of the three healthy donors. During fermentations, samples were collected daily from the fermentative medium and the atmospheric phase for downstream analyses.

Five bioreactors were inoculated with the same fecal suspension and ran in parallel for 28 days. No treatment was applied in control condition (control, red bioreactor). After a 6-day period of microbiota stabilization, the four other bioreactors were treated with 500 µg/mL ciprofloxacin for 6 days to induce gut microbiota dysbiosis. Out of the four reactors, one received no FMT treatment (ATB control, blue fermenter). After a 2-day period of antibiotic wash-out, FMT treatment was performed in the three last bioreactors with different modes of administration: 30 g enema (green bioreactor), 10 g enema (pink bioreactor) or oral capsule (grey bioreactor). Enema treatments were administered at day 14 and day 15. Oral capsules were administered three times per day for seven days from day 14 to day 20. The recovery period was defined as days of fermentation after cessation of FMT treatment. Experiments were performed in triplicate with feces from three different healthy adult donors (Run1, Run2, and Run3).

### 2.4. Antibiotic Dosage

Ciprofloxacin concentrations in the fermentative medium were determined using a TurboFlow^TM^ technology (TLX) coupled to Liquid Chromatography-High Resolution Mass Spectrometry (LC-HRMS, ORBITRAP^®^ technology) at the pharmacological and toxicological analytic unit (CREPTA) of Clermont-Ferrand university hospital, using a method adapted from Hösl et al. 2018 [32] and Lefeuvre et al. 2017 [33].

### 2.5. Gut Microbiota Activity

#### 2.5.1. Gas

Analysis of O_2_, N_2_, CO_2_, CH_4_, and H_2_ present in the bioreactor atmospheric phase was performed using a 490 Micro-gas chromatograph (Agilent Technologies, Santa Clara, CA, USA) equipped with two columns, Molecular Sieve 5A, and PoraPlot U, coupled with TCD detectors. Argon was used as the carrier gas. Gas composition was determined using calibration curves made from ambient air (78% N_2_, 21% O_2_, 0.04% CO_2_) and two gas mixtures A (5% CO_2_, 5% H_2_, 90% N_2_) and B (20% CO_2_, 80% H_2_). Results were expressed in relative percentages. Total volume of gases overproduced per day (in mL) was also measured by connecting a gas bag to each bioreactor.

#### 2.5.2. Short Chain Fatty Acids (SCFAs)

Samples collected from fermentative medium were centrifuged at 18,000 g for 15 min at 4 °C and supernatants were filtered (0.45 µm). Concentrations of the three main SCFAs (acetate, propionate and butyrate) were determined using high performance liquid chromatography (HPLC) (Elite LaChrom, HITACHI, San Jose, CA, USA) coupled with a diode-array detector. The HPLC column (Concise Separations, San Jose, CA, USA, ICE-99-9865) and its guard column were maintained at 50 °C. Sulfuric acid 0.008 N was used as mobile phase and SCFAs were separated at a flow rate of 0.6 mL/min. Data was analyzed by the EZChrom Elite software at 205 nm. SCFAs concentrations (expressed in mM or relative percentages) were calculated from standard curves established with known concentrations of acetate, propionate and butyrate (0, 10, 25 and 40 mM).

### 2.6. Gut Microbiota Composition

#### 2.6.1. Flow Cytometry Analysis

Concentrations of viable bacteria in the fermentation medium were determined by flow cytometry through a live/dead analysis. Samples were 10-fold diluted in sterile physiological water to reach 10^−4^ dilution factor. Bacteria were double-stained with the green-fluorescent DNA SYTO 9 dye labelling all bacteria and the red-fluorescent Propidium Iodide dye only penetrating and staining cells with damaged membranes (Molecular probes, Eugene, OR, USA, L34856). Bacterial suspensions were thus incubated for 15 min at room temperature in the dark with 3.3 mM SYTO 9 and 0.375 mM Propidium Iodide and transferred into BD Trucount™ Tubes (BD Biosciences, San Jose, CA, USA). Flow cytometry analysis was performed on a BD™ LSR II cytometer (BD Biosciences, San Jose, CA, USA) and data were collected with BD FACSDiva^TM^ software. Gating on forward-angle light scatter/side-angle light scatter was used in order to differentiate bacteria from the background. Then, combined red and green fluorescence dot-plots were used to distinguish the various subpopulations. Results were expressed as viable cells per mL of fermentative medium.

#### 2.6.2. qPCR Analysis

Total DNA was extracted from fermentative medium using SmartExtract-DNA Extraction Kit (Eurogentec, Liège, Belgium, SK-DNEX-100), according to manufacturer’s instructions. DNA quantity was evaluated with a NanoDrop^TM^ 2000 (Thermo Scientific, Wilmington, DE, USA). Samples were stored at −20 °C prior to analysis. Total bacteria concentration was quantified by qPCR analysis performed on a Stratagene Mx3005P apparatus (Agilent, Waldbronn, Germany) using Takyon Low ROX SYBR 2X MasterMix blue dTTP kit (Eurogentec, Liège, Belgium, UF-LSMT-B0701). Each reaction was run in duplicate in a final volume of 10 µL with 5 µL of Master Mix, 0.45 µL of each primer (10 µM), 1 µL of DNA sample (10 ng/µL), and 3.1 µL of ultra-pure water. The following primers targeting 16S rRNA gene were used: BAC338R, 5′- ACTCCTACGGGAGGCAG-3′, and BAC516F, 5′-GTATTACCGCGGCTGCTG-3′, as described by Yu and colleagues [34]. The amplification conditions consisted in 1 cycle at 95 °C for 5 min followed by 40 cycles of 95 °C for 30 s, 58 °C for 30 s, and 72 °C for 30 s. A final cycle of 5 min at 95 °C was included. Standard curves were generated from genomic DNA extracted from ARCOL samples, standardized to 10 ng/μL and serially diluted from 10^10^ to 10^0^ gene copy/mL. Final results were expressed as copy/mL of genomic DNA.

#### 2.6.3. 16S rRNA Gene Sequencing and Bioinformatics Analysis

Genomic DNA was extracted using the NucleoSpin Soil kit (Machery Nagel, Düren, Germany, 740780.50) and samples stored at −20 °C before analysis. 16S rRNA gene sequencing was performed by Eurofins Genomics (Ebersberg, Germany). A sequencing library targeting the V3-V4 region of the 16S rRNA gene was constructed for each sample using the MyTaq HS-Mix 2X (Bioline, Memphis, TN, USA, BIO-25045) according to manufacturer’s instructions. Libraries were then pooled in an equimolar mixture and sequenced in paired-end (2 × 300 bp) MiSeq V3 runs, Illumina. After amplicon merging using FLASH [35] and quality filtering using Trimmomatic [36], host sequence decontamination was performed with Bowtie2 [37]. Operational Taxonomic Unit (OTU) sequence clustering was performed with an identity threshold of 97% using VSEARCH [38] and taxonomic profiling was then performed with the Silva SSU database Release 128 [39]. Taxonomic and diversity analyses were performed with R Statistical Software (R Core Team 2015, version 3.4.4) [40] using vegan and phyloseq packages. For fair comparison, the sequence number of each sample was randomly normalized to the same sequencing depth i.e., 50,000 amplicons per sample and normalized by total bacteria count based on qPCR results. Diversity measures correspond to the median value of 20 subsamplings per sample.

All sequencing data were deposited to the National Center for Biotechnology Information Sequence Read Archive under accession number PRJNA642894.

### 2.7. Dysbiosis Criteria

Criteria selected to determine microbial dysbiosis periods in ARCOL system were based on modifications of both gut microbiota activity and composition compared to stabilized conditions. For microbiota activity, the following parameters were selected: redox potential values, NaOH consumption, total gas production, CO_2_ concentration, and SCFA concentrations. Regarding gut microbiota composition, the selected parameters were the following: total viable bacteria as determined by flow cytometry, total bacterial populations measured by qPCR, richness, Shannon and Bray Curtis indexes. In order to establish a dysbiotic period, each day of fermentation from day 6 for a treated bioreactor was compared to day 6 (corresponding to the end of stabilization phase) of the same bioreactor for all 16S rRNA gene analysis-related criteria (abundance and diversity indexes). For all other criteria, each day of fermentation from day 6 for a treated bioreactor was compared to the same day of the control bioreactor. All selected criteria and threshold values are summarized in Table 2.

### 2.8. Statistical Analysis

SCFAs (in mM), cytometry, and qPCR data were analyzed using a one-way repeated measure analysis of variance (ANOVA) followed by a Newman-Keuls multiple comparisons test. The statistical analysis was performed using GraphPad Prism software 8.0 (GraphPad Software, Inc., San Diego, CA, USA). Results were expressed as means ± SEM (*n* = 3). Differences were considered statistically significant when *p* < 0.05.

## 3. Results

### 3.1. Monitoring of In Vitro Fermentations

#### 3.1.1. NaOH Consumption

Microbial fermentation activities lead to organic acid production such as SCFA and to a subsequent pH decrease in the bioreactor, resulting in NaOH consumption to maintain the pH at its set point value. Whatever the bioreactor, NaOH consumption was stable before ATB treatment. Addition of ciprofloxacin led to an immediate interruption in NaOH consumption that persisted 3 to 4 days after the end of ATB treatment, depending on the experimental runs. FMT treatments led to an earlier restart of NaOH consumption compared to ATB control, two days before for 30 g and 10 g enema and only one day before for oral capsule (in two out of the three replicates, Run3 being similar to ATB control).

#### 3.1.2. Redox Potential

Redox potential was also evaluated as an indicator of fermentation activities and anaerobiosis. Before ATB treatment, redox potential stabilized at around −400 mV in all bioreactors. Addition of ciprofloxacin led to an immediate change with a sharp increase in redox values (up to 0 mV). At the end of ATB treatment, redox potential slowly decreased to reach baseline values of stabilization phase within 3 to 8 days depending on replicates. Enema treatments (both 30 and 10 g) allowed an earlier return to baseline values (except for Run3), four days before ATB control in Run1 and two days before in Run2. For capsule, a donor dependent effect was observed with an earlier return to baseline in Run1 (four days before ATB control), no effect in Run2 and a slower return to a stabilized state compared to ATB control in Run3.

### 3.2. Gas Production

At the end of stabilization phase (day 6), gas composition of the atmospheric phase was the same in all tested conditions with approximately 95% of CO_2_, 4% N_2_, 1% H_2_, and less than 1% O_2_ (Figure 2a–e). This result confirms the ability of maintaining anaerobiosis inside bioreactors without flushing with CO_2_ or N_2_ during the total course of fermentation. These relative percentages remained constant throughout control experiments (Figure 2a). As for NaOH consumption, addition of ciprofloxacin led to an immediate termination in gas overproduction that persisted four days after the end of ATB treatment. This resulted in negative pressure in fermenters that required N_2_ injection into bioreactors. Moreover, ATB treatment was associated with a change in gas composition (Figure 2b–e). As illustrated, a sharp increase in relative percentages of N_2_ was observed due to flushing (from 55 to 80%) as well as a lower but clear increase in H_2_ (1–20%) and O_2_ (5–10%) that cannot be linked to any gas leak (connection of a gas bag filled with N_2_). Consequently, CO_2_ relative percentages decreased to 15–20% (Figure 2b–e). For ATB control, a return to baseline profiles was observed only 10 days after the end of ciprofloxacin treatment, i.e., at day 22 (Figure 2b). When FMT treatments were applied (Figure 2c–e), gas production restarted faster (1 or 2 days before ATB control) for all modes of administration, except for oral capsule in Run3. Recovery of stabilized profiles (similar to that observed at day 6) was also obtained earlier for 30 and 10 g enema treatment (day 19) or capsule (day 21) compared to ATB control (day 22).

### 3.3. SCFA Production

Whatever the fermentations, SCFA profiles stabilized at day 6 with relative percentages around 60%, 25%, and 15% for acetate, propionate, and butyrate, respectively (Figure 2f–j) and a total concentration of 130–140 mM (Figure 2k). In the control experiment, these percentages remained similar throughout fermentation (Figure 2f). Addition of ciprofloxacin induced a significant decrease (*p* < 0.05) in total SCFA concentrations to reach an approximate concentration of 50 mM at the end of ATB treatment (Figure 2k). ATB treatment also induced changes in SCFAs profiles (Figure 2g–j) with an important increase in relative percentages of propionate (80–95% at day 12) associated with a decrease in acetate (5–15%), and butyrate (0–5%). These changes persisted at the end of ATB treatment since total SCFA concentrations returned to baseline within 8 days (day 20, Figure 2k). Thus, two additional days (day 22) were needed to recover SCFA proportions similar to those observed at the end of stabilization (Figure 2g). FMT treatment led to a sharp increase in total SCFA concentrations at day 15 for both enema treatments (around 280 mM), i.e., immediately after the first injection (Figure 2k). A donor-dependent response can explain this high variability with a peak at day 15 for Run1, such effect is 4 to 5 times higher than the one observed for Run2 and Run3. Similarly, but to a lesser extent, FMT capsule led to an increase in total SCFA concentrations to reach a maximum of 150 mM at day 17 (Figure 2k). All FMT treatments led to a clear reduced time needed to return to stabilized state (2–4 days before ATB control), with both total SCFA concentrations (Figure 2k) and proportions (Figure 2h–j) similar to baseline values at day 18. Regarding oral capsule, stabilization of total SCFA concentrations occurred at a lower level (100 mM versus 130–140 mM).

### 3.4. Quantification of Total Bacteria

In control experiments, total bacteria stabilized at around 6–7 × 10^9^ 16S rRNA gene copies/mL, as determined by quantitative PCR analysis (Figure 3a to 3d). Microbial populations were mostly composed of viable cells since a similar population level was obtained by flow cytometry (Figure 3e–h). Total bacteria number and viable bacteria amount were both significantly impacted by ATB treatment with a regular decrease until day 12 in most bioreactors (up to 3–4 logs as demonstrated by qPCR and 2–3 logs as demonstrated by flow cytometry). Nevertheless, the influence of ciprofloxacin impact was not similar in all the experiments, with a minor influence on total and viable bacteria in Run2 for enema 30 g and capsule conditions (Figure 3b–f). On average, the return to baseline for ATB control condition occurred progressively within 6 and 7 days after the end of ciprofloxacin administration for total and viable bacteria, respectively (Figure 3d–h). Likewise, all FMT treatments enable a return of viable bacteria concentrations to stabilized values 3 days before ATB control condition (*p* < 0.05, Figure 3h). Nevertheless, an important variability was observed between the three replicates, more particularly for 30 g enema and capsule conditions. Especially, the time to recover initial levels in Run3 was similar for FMT capsule when compared to ATB control (Figure 3c–g).

### 3.5. Gut Microbiota Structure

#### 3.5.1. Composition of Initial Fecal Inoculum and Following Stabilization in ARCOL

Sequencing analysis of the initial fecal suspensions at the phylum level indicated that donor 1 and donor 2 (Appendix A, D0, Run1 and 2) exhibited microbial profiles mainly composed of *Firmicutes* (70% and 65%, respectively) and *Bacteroidetes* (30% and 35%, respectively). Donor 3 (Appendix A, D0, Run3) had a reverse *Firmicutes*/*Bacteroidetes* ratio (35%/60%) with a higher prevalence of *Proteobacteria* (3%). At the family level, dominant taxa of fecal suspensions were as follows: *Ruminococcaceae* (40%), *Bacteroidaceae* (25%)*, Lachnospiraceae* (20%), and *Veillonellaceae* (7%) for donor 1 (Figure 4, Run 1); *Ruminococcaceae* (30%), *Prevotellaceae* (20%), *Lachnospiraceae* (20%), *Bacteroidaceae* (10%) and *Veillonellaceae* (10%), for donor 2 (Figure 4, Run2); and *Bacteroidaceae* (50%), *Ruminococcaceae* (25%), *Lachnospiraceae* (10%), and *Rikenellaceae* (%) for donor 3 (Figure 4, Run3).

At the end of stabilization phase in ARCOL (Appendix A, day 6) phyla profiles were quite similar for all runs with a large majority of *Bacteroidetes* (60–70%), followed by *Firmicutes* (20–40%), and *Proteobacteria* (1–3%). At the family level (Figure 4), profiles at day 6 appear to be run-dependent with close composition for Run1 and Run3 while being clearly different for Run2. Run1 and Run3 exhibited a high abundance of *Bacteroidaceae* (70–80% and 65–75%, respectively) followed by *Ruminococcaceae* (6% and 15%, respectively) and *Lachnospiraceae* (5% and 10%, respectively). Gut microbiota in Run1 also displayed 4% of *Veillonellaceae.* For Run2, profiles are composed of stable abundances of *Ruminococcaceae* (30–40%), *Lachnospiraceae* (10–15%), and *Veillonellaceae* (1–3%) while *Prevotellaceae* and *Bacteroidaceae* were more variable between conditions (from 2 to 40%). Stabilized profiles remain almost constant for all control experiments during the 28 days of fermentation (Figure 4).

Experiments were performed as described in Figure 1 in triplicate with fecal samples from three healthy adult donors (Run1, Run2 and Run3). Different conditions were applied: no treatment (control), ciprofloxacin (ATB control), ciprofloxacin and 30 g enema, ciprofloxacin and 10 g enema, and ciprofloxacin with oral capsules. Microbial composition was determined at the family level by 16S rRNA gene sequencing and expressed as relative abundances.

#### 3.5.2. Impact of ATB Treatment

Addition of ciprofloxacin led to pronounced changes in microbial profiles even at the phylum level. For Run1 and Run3, at the phylum level, major changes were characterized by an increase in *Firmicutes* abundance and a disappearance of *Proteobacteria* (Appendix A). In Run1 only, *Verrucomicrobia* (up to 3%) appeared during ATB treatment. For Run2, opposite trends were observed with a decrease in *Firmicutes* abundance associated with a sharp increase in *Proteobacteria* (up to 90% in ATB control). At the family level, ATB treatment had also a strong influence on microbial structure with variations between experimental runs and even between bioreactors for a single run. Major changes for Run1 were represented by an increase in *Ruminococcaceae* and *Lachnospiraceae* together with a loss of *Veillonellaceae* and *Porphyromonodaceae* (Figure 4). For Run2, *Prevotellaceae* and *Alcaligenaceae* abundances were mostly increased while those of *Bacteroidaceae* and *Porphyromonodaceae* declined. Lastly for Run3, according to bioreactors, the main variations observed were a bloom of one or several families among the *Enterococcaceae, Planococcaceae, Clostridiaceae* or *Lachnospiraceae*. Perturbations of microbial profiles persisted following cessation of ATB treatment in control experiments (ATB control). Stabilization occurred only around day 25–27 but with different profiles when compared to day 6, both at phylum (Appendix A) and family (Figure 4) levels.

#### 3.5.3. Effect of FMT Treatments

All FMT treatments induced a rapid and clear shift (from day 15) in microbial profiles both at phylum (Appendix A) and family levels (Figure 4). Microbial abundances kept evolving after this initial shift until 5–8 days after the first FMT administration to reach a new stable profile close from the one observed at the end of stabilization phase. Some taxa, such as *Veillonellaceae* in all runs, *Bacteroidales* S24-7 group in Run2 and *Alcaligenaceae* and *Porphyromonodaceae* in Run3, which disappeared during ATB treatment, have been shown to reappear in FMT-treated bioreactors only (but not in ATB control). On the contrary, some families present in control experiments, such as *Prevotellaceae* in Run3, were no longer present after ATB treatment even in FMT-treated bioreactors (Figure 4). Interestingly, in Run2, *Prevotella 7,* the main genus from *Prevotellaceae,* present at the end of stabilization phase was substituted by *Paraprevotella* during ATB treatment which disappeared again in favor of *Prevotella 7* when capsule and 30 g enema treatments were applied.

### 3.6. Microbial Richness and Diversity

#### 3.6.1. α-. Diversity

Alpha-diversity was first evaluated by calculating sample richness at the OTU level, i.e., the number of different OTUs reflecting species in a sample (Figure 5, panel A). In control experiments, for all three donors (Run1, Run2 and Run3), richness stabilized at around 150–200 OTUs all along the fermentation process. Administration of ciprofloxacin led to a rapid and pronounced decrease of richness index to reach less than 10 OTUs at the end of ATB treatment. At the end of ciprofloxacin injection in ATB control experiments, richness increased to stabilize at day 18–20, but at lower values when compared to control assays (around 80–100 OTUs). Interestingly, FMT treatment enable a return of richness values to baseline levels within 6-8 days after the end of ATB treatment. Shannon index was also calculated to better picture diversity and species distribution in the various samples (Appendix A). In control experiments, Shannon index stabilized at values around 2–3 from day 6 to day 28. ATB treatment was associated with a sharp decrease of Shannon index, with value below 1. For ATB control, Shannon index regularly increased to reach baseline levels the last 2 days of experiments, with an exception for Run3 where stabilized values remained lower than the one at day 6 (around 2 versus 3). When FMT treatments were performed, Shannon index recovered baseline values within 2 to 6 days after the first administration for enema formulae and after 6 days for oral capsule.

Experiments were performed as described in Figure 1 in triplicate with fecal samples from three healthy adult donors (Run1, Run2 and Run3). Different conditions were applied: no treatment (control, red), ciprofloxacin (ATB control, blue), ciprofloxacin and 30 g enema (green), ciprofloxacin and 10 g enema (purple), and ciprofloxacin with oral capsules (black). Richness (panel A) and Bray-Curtis (panel B, compared to day 6 values) indexes were determined at the OTU level after 16S-rRNA gene sequencing. Bray-Curtis values represent similarity of samples versus day 6.

#### 3.6.2. β-. Diversity

Bray-Curtis dissimilarity was calculated as an indicator of β-diversity compared to values obtained at day 6 (end of stabilization phase for each bioreactor). Results obtained for the three replicates Run1, Run2 and Run3 are presented in Figure 5 panel B. Whatever the replicates, similar trends were observed. For control experiments, Bray-Curtis similarity index versus day 6 remained stable during fermentations (except for Run3 where a slight decrease was noted). When no FMT treatment was applied to bioreactor following ciprofloxacin administration (ATB control), a clear shift from control experiments was observed with values being much lower (from 0.2 to 0.3). Nonetheless, whatever the replicates (Run1, Run2 or Run3), FMT treatment induced a clear restoration of microbial diversity since all samples progressively normalize to control values, except for 10g enema and oral capsule in Run2 (indexes at around 0.5 at the end of fermentation). For all other tested conditions, return to baseline values was observed between day 18 and day 20 for all FMT modes of administration.

### 3.7. Determination of Dysbiotic Periods

In order to assess FMT efficiency in restoring gut microbiota composition and activity, as well as the influence of mode of administration, the number of “dysbiotic days” was determined. This number of “dysbiotic days” was calculated for each variable listed in Table 2 and for each tested condition (ATB control, 30 g enema, 10 g enema, and capsule). Results were expressed as average of the three replicates (Appendix A) and represented in Figure 6. These results confirmed that ATB control experiments exhibited the highest number of dysbiotic days with an average of 12.1 days (Appendix A). FMT treatment clearly decreased dysbiotic periods with a similar value of 7.6 days for both 30 g enema and 10 g enema. For oral capsule, a longer dysbiotic period of 8.3 days was observed (non-significant). When analyzing results in depth especially at microbiota structure and activity levels, different situations were observed depending on FMT mode of administration. Regarding microbial activity, capsule was the less efficient technique with 9.4 days of dysbiosis, followed by 10 g enema (8.7 days), while 30 g enema showed the best score (7.7 days). This difference was mainly due to a high number of dysbiotic days for capsule in relation with acetate production (Figure 6) and probably results from a latent period required for microbial revivification of capsule freeze-dried form. Regarding microbial structure, different ranking was observed, with 30 g enema being the less efficient (7.6 days) followed by capsule (6.9 days) and 10 g enema (6.1 days). The lower efficiency of 30 g enema resulted from a higher number of dysbiotic days in relation with all the microbial diversity indexes.

Experiments were performed as described in Figure 1 and different conditions were applied: ciprofloxacin (ATB control, blue), ciprofloxacin and 30 g enema (green), ciprofloxacin and 10 g enema (purple), and ciprofloxacin with oral capsules (black). For each tested condition, the number of dysbiotic days was determined and compared to the appropriate control using criteria described in Table 2 (in relation with both gut microbiota structure and activity) and expressed as mean number of dysbiotic days (*n* = 3).

## 4. Discussion

In this study, we described for the first time the use of a dynamic computer-controlled in vitro model of the human colon, namely ARCOL, as an alternative to in vivo assays in rodents for testing different FMT formulations following antibiotic treatment. The in vitro model was challenged with ciprofloxacin, an antibiotic widely used in humans, to reproduce clinically relevant gut microbiota perturbations. Then, the “dysbiotic” model was efficiently used to evaluate different modes of FMT administration. This study demonstrates that a new oral capsule was as efficient as more traditional invasive enemas to restore gut microbiota structure and activity and to decrease the number of disturbed days.

The first aim of the present study was to establish a medically relevant “dysbiotic” in vitro model of the human gut ecosystem. For this purpose, we used the in vitro colon model ARCOL, set-up to mimic, based on in vivo data, not only physicochemical parameters of the adult human colon but also microbial components [27,28,29,30,41]. This model was previously validated under healthy conditions, i.e., with a non-disturbed microbiota. As the drastic effect of antibiotics on human gut microbiota is well acknowledged by the scientific community [42], we decided to apply an antibiotic treatment to induce gut microbiota perturbations in ARCOL system, as already performed in other colon models [25,26,43]. Ciprofloxacin, a fluoroquinolone antibiotic widely used in hospital settings to treat various digestive and extra-digestive bacterial infections, was chosen in the present study [44]. This antibiotic was also selected due to available information on its pharmacokinetic in humans, especially on its fecal excretion [45]. The dose (500 µg/mL) and the mode of administration (first initial input followed by a continuous supply for 7 days) were established considering hospital practices, ciprofloxacin metabolism (percentage of absorption in the human upper gut and estimated fecal antibiotic clearance) and in vitro colon parameters (total volume and retention time). Ciprofloxacin concentrations were checked by antibiotic dosage throughout ARCOL experiments (500 µg/mL). Our results clearly indicated that ciprofloxacin induced in vitro rapid changes in gut microbial composition and activity that heightened during antibiotic administration. These modifications clearly include a reduction of overall microbial richness and diversity and a loss in microbial fermentation capacities, in accordance with in vivo data in humans, even if available data have not been obtained exactly under the same operational conditions (e.g., dose) [46,47]. In addition, in some replicates, a rise in dominance of bacterial species usually subdominant, including pathobionts such as *Enterobacteriaceae* and some *Clostridiaceae,* was observed. All these changes are indicative of a gut microbiota dysbiosis state in accordance with human data [4,5]. Interestingly, at the family level, effects of ciprofloxacin were widely donor dependent in ARCOL, as previously reported in humans and mice [48]. In the current in vitro study, we describe for the first time an original index that quantitates the dysbiosis state in ARCOL. This dysbiosis index was calculated based on the total number of dysbiotic days in bioreactors. To consider all aspects related to dysbiosis definition [4,5], this index integrates criteria related to both microbial structure (bacterial population relative abundances, microbial richness and diversity) and functions (related to main fermentation products such as gases and SCFA).

Once the “dysbiotic” in vitro colon model was established and validated according to in vivo data in humans from literature, we evaluated the ability of various FMT formulations to restore the balance of gut microbial communities in ARCOL model. Three different forms were tested, i.e., two dosages of enema and one oral capsule. Until 1990, enema was the method of choice for FMT and still remains frequently used in hospitals [14]. Enema is less invasive, easier to perform and relatively less expensive than colonoscopy or upper gastro-intestinal routes, even if the approach can be limited by patient compliance and difficulty to retain the suspension. Two doses were tested, 30 g enema which is commonly used in clinical practices and 10 g enema to comparatively assess the efficiency of a lower microbial quantity. The effect of these two formulations was compared to that of a new caecum-releasing capsule containing a freeze-dried form of the enema formulation. Oral capsule is the most recently developed mode of stool delivery, the first formulation being described in 2014 [49,50]. Capsules which are esthetically pleasant, convenient, and minimally invasive are preferred by patients and are advantageously the cheapest FMT mode of administration [19,49,50]. To be as close as possible to clinical procedure, enemas were directly introduced into ARCOL model, since in humans they are delivered into the distal colon via a cannula. The oral capsule tested in this study was specifically designed to open and release its content at the end of human small intestine only. In a preliminary pilot study, we evaluated capsule integrity in ileal effluents of the human TNO gastrointestinal, TIM, system by visual control and bacterial numeration [14]. As the capsules remained intact until the end of small intestine in vitro, capsule content was directly added as a suspension in ARCOL system in the present work.

Our data demonstrated that all tested forms were able to accelerate return to a stable state compared to ATB control, thereby reinforcing the resilience of the ecosystem. FMT treatments had an immediate impact on gut microbiota structure while there was a latency for recovery of a stable microbial activity. This phenomenon was due to necessary adaptation of newly added bacteria. Interestingly, the lowest enema dosage (10 g) was as efficient as the highest one (30 g) and reduced in a similar way the number of dysbiotic days (of approximately 4.5 days). To our knowledge, no previous published report had already investigated a dose effect for FMT enema in vitro. These in vitro results suggest that bacterial load may be reduced without any deleterious effect while restoring microbial balance after antibiotic-induced dysbiosis. The oral capsule proved almost as efficient as enema forms (–3.8 dysbiotic days), despite a lower amount of administered bacteria (100 times lower) and a longer administration period (1 week compared to two days). In previous studies, two approaches were mainly developed for FMT oral capsules, first freezing at −80 °C with glycerol and more recently freeze-drying with various cryoprotectants, as performed in the present study [50,51,52]. Interestingly, Jiang and colleagues showed in mice that there was no difference in efficiency between frozen and lyophilized capsules in rCDI treatment. Moreover, the authors demonstrated that products can be stored up to 7 months without losing gut microbiota composition and therapeutic efficacy [51]. Oral capsules were also designed for delivery in various luminal segments of the gastro-intestinal tract. A comparative study between two capsules preparations, with either gastric or colonic delivery properties, showed that the colonic-release form tended to be the most effective in rCDI, particularly in restoring *Bacteroidetes* phylum and increasing gut microbial diversity [50]. This result is in accordance with the strategy promoted in the current study testing a caecum-release oral formulation. To date, few studies in humans have already compared the efficiency of oral capsules (frozen or lyophilized) to more traditional FMT routes (such as colonoscopy or enema) for the treatment of rCDI. In accordance with our results, despite the differences of tested formulations from other studies and ours, available data in humans indicate that oral capsules are as effective as traditional modes of administration, especially to restore bacterial diversity [19,52]. Allegretti and colleagues have shown in humans that the lower dose treatment with colonic release capsules (10 capsules in a single administration) was equally effective to cure rCDI than the higher treatment-dose (30 capsules) [50]. This is fully in accordance with our results on enema forms, showing the relevance of in vitro approach for FMT evaluation.

This study confirms the potential of in vitro gut models to mimic gut microbiota dysbiotic state and for subsequent assessment of FMT efficiency. Such approach appears as a relevant alternative to in vivo animal assays in preclinical phases. Interestingly, in vitro results could appropriately document technical files ahead of clinical trials in humans. Although they accurately report the ecological dynamics in terms of structure and function, these models are hindered by their inability to fully reproduce the overall processes occurring in vivo, especially hormonal and nervous controls, feedback mechanisms, local immune system and host-bacterial mutualism. Particularly, patients receiving FMT are often subjected to diverse external stresses that cannot be integrated in vitro but may widely influence gut physiology and microbiota restoration. Yet, for obvious regulatory, ethics and cost reasons, in vitro colon models are advantageous over in vivo assays due to their flexibility, accuracy and reproducibility [53]. This approach is also adequately in line with the European and North American 3R rules aiming to minimize the number of animals used for research purpose and promote the development of alternative in vitro methods. As a stable microbiome can be maintained in bioreactors over a long timeframe, the effect of successive treatments can also be tested. In addition, gut models can be adapted to perform colon-segment specific research [31,54] and several bioreactors can be inoculated with the same fecal sample to perform in parallel control and treatment experiments. Lastly, as shown here, inter-individual gut microbiota variability can be considered by performing replicates with fecal samples from different donors. In the present study, ARCOL was inoculated with stools from healthy adult donors but we can envision to extend its potentialities (as previously performed in other human colon models) by using fecal samples from selected age groups, such as infants [55,56,57] or elderly subjects [58,59], or from patients suffering from cancers, inflammatory bowel diseases or metabolic disorders [60,61,62,63]. To be fully relevant, a complete adaptation of physicochemical parameters of the colon (pH, transit time, composition of nutritive medium and biliary salts) to the specific targeted population would be required. This would unlock new applications in human health through in vitro evaluation of FMT potential to restore gut microbiota eubiosis in these pathological situations.

To conclude, this study describes for the first time the use of an in vitro human colon model, adapted to reproduce clinically relevant perturbations by antibiotherapy, for the evaluation of various FMT formulations. By integrating the main physicochemical parameters of the human colon (pH, retention time, nutrient supply and anaerobiosis), ARCOL model was shown to capture microbial diversity and inter-individual variability of the human gut microbiome. Treatment with ciprofloxacin led to a marked state of dysbiosis with a sharp decrease in fermentation activities, a loss of microbial diversity and a shift in bacterial populations, as previously described in vivo in humans. The three formulations tested for autologous FMT, i.e., 30 g enema, 10 g enema and a new caecum-release oral capsule, showed similar effects by clearly reducing the time to restore gut microbiota structure and activity. Due to regulatory, ethic, cost and technical advantages, in vitro gut models such as ARCOL can be advantageously used in preclinical phases as an alternative to in vivo assays in animals. Their potential in human health may be extended to the evaluation of allogenic FMT and could help in selecting the best clinical protocols (e.g., period for treatment, frequency and duration of FMT administration) when targeting various age groups and intestinal or extra-digestive pathologies.

## Figures and Tables

**Figure 1 microorganisms-09-00358-f001:**
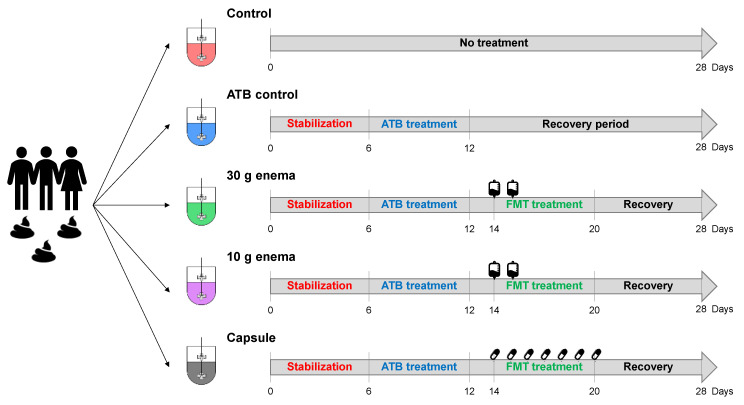
Experimental design of in vitro fermentations in ARtificial COLon (ARCOL) system.

**Figure 2 microorganisms-09-00358-f002:**
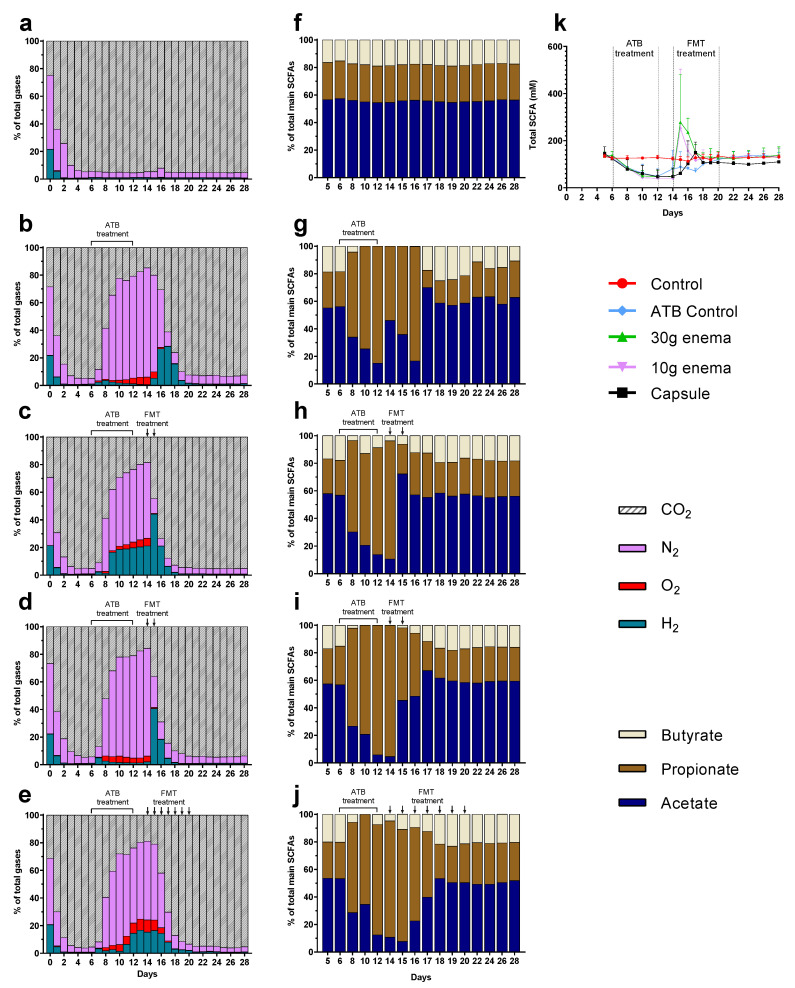
Effect of fecal microbiota transplantation (FMT) treatment on gas and short chain fatty acid (SCFA) production. Experiments were performed as described in Figure 1 and different conditions were applied: no treatment (control, subfigures (**a**) and (**f**)), ciprofloxacin (ATB control, subfigures (**b**) and (**g**)), ciprofloxacin and 30 g enema (subfigures (**c**) and (**h**)), ciprofloxacin and 10 g enema (subfigures (**d**) and (**i**)), and ciprofloxacin with oral capsules (subfigures (**e**) and (**j**)). Gas composition was determined by gas chromatography and results expressed as mean relative percentages (subfigures (**a**–**e**), *n* = 3). Main SCFAs (acetate, propionate and butyrate) were analyzed by high performance liquid chromatography. Results were either expressed as mean relative percentages (subfigures (**f**–**j**), *n* = 3) or as total SCFA concentrations ± SEM (in mM, subfigure (**k**)).

**Figure 3 microorganisms-09-00358-f003:**
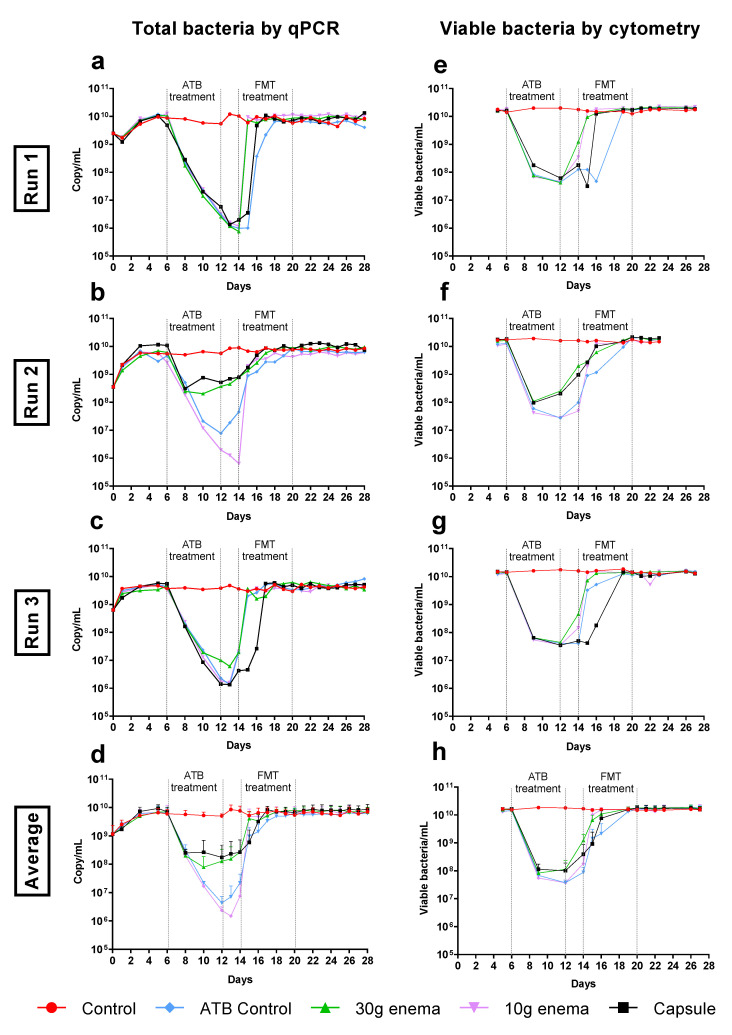
Effect of fecal microbiota transplantation (FMT) treatment on total bacteria. Experiments were performed as described in Figure 1 in triplicate with fecal samples from three healthy adult donors (Run1, Run2 and Run3). Different conditions were applied: no treatment (control, red), ciprofloxacin (ATB control, blue), ciprofloxacin and 30 g enema (green), ciprofloxacin and 10 g enema (purple), and ciprofloxacin with oral capsules (black). Total bacteria was determined by qPCR analysis and expressed as numbers of 16S rRNA gene copies/mL in Run1 to Run3 (subfigures (**a**–**c**)) or in mean number of copies/mL ± SEM (subfigure (**d**), *n* = 3). Total viable bacteria was determined by flow cytometry through a live/dead analysis and expressed as number of viable cells/mL in Run1 to Run3 (subfigures (**e**–**g**)) or in mean viable cells/mL ± SEM (subfigure (**h**), *n* = 3).

**Figure 4 microorganisms-09-00358-f004:**
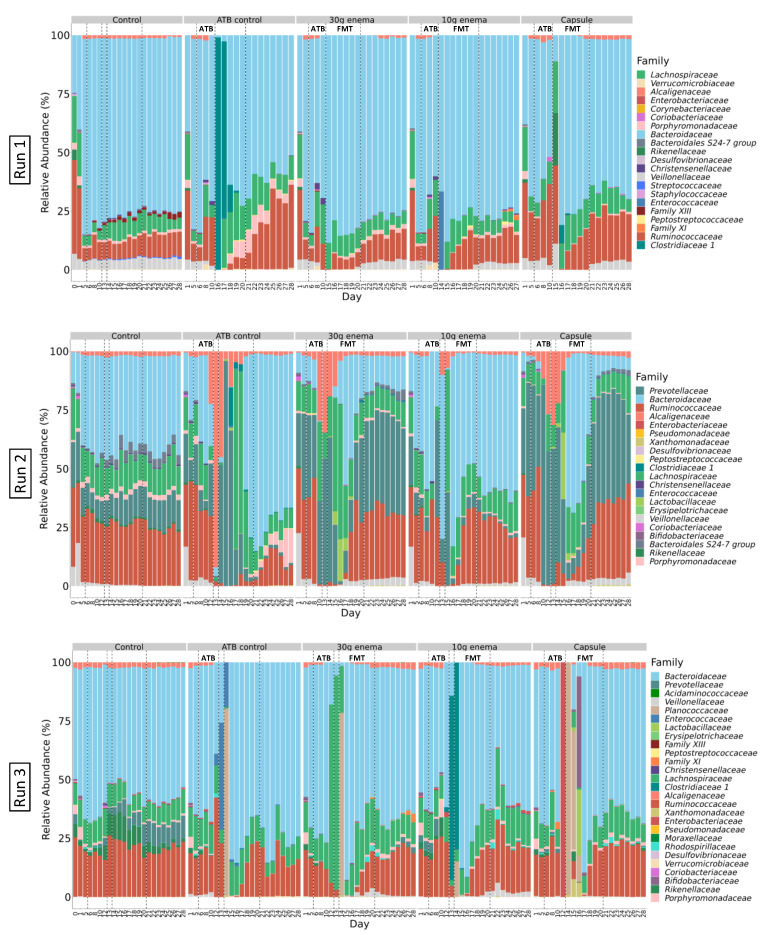
Effect of fecal microbiota transplantation (FMT) treatment on microbial composition at family level.

**Figure 5 microorganisms-09-00358-f005:**
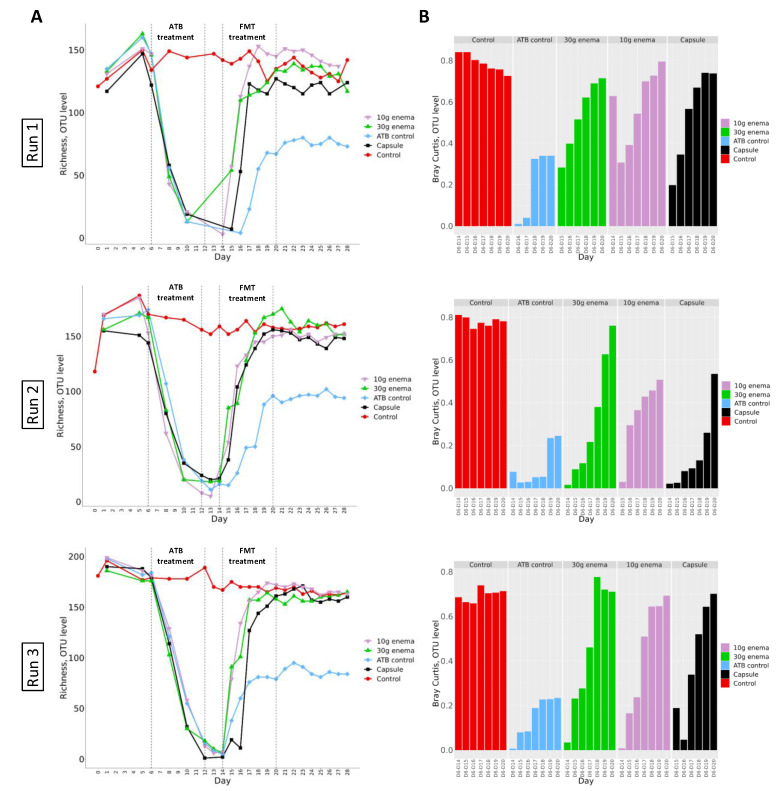
Effect of fecal microbiota transplantation (FMT) treatment on richness and Bray-Curtis indexes at OTU level.

**Figure 6 microorganisms-09-00358-f006:**
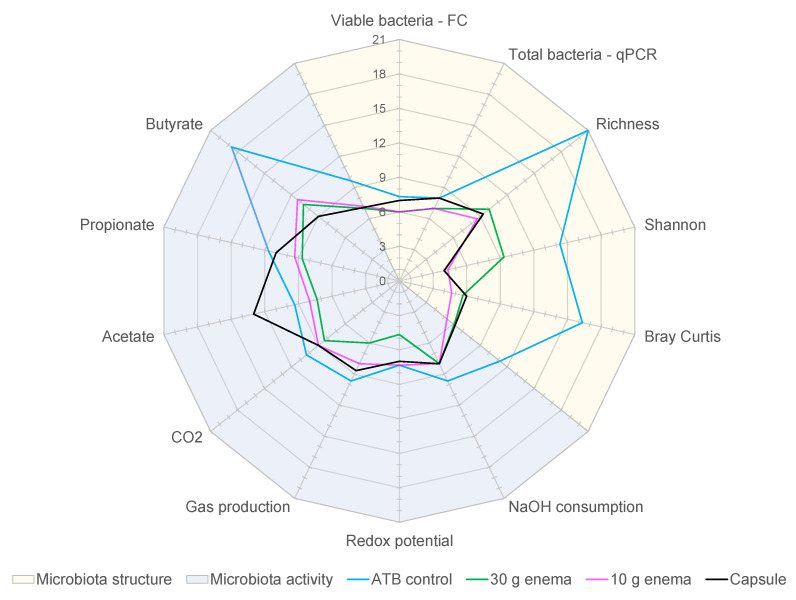
Effect of fecal microbiota transplantation (FMT) treatment on the duration of dysbiosis.

**Table 1 microorganisms-09-00358-t001:** Composition of the nutritive medium used to feed in vitro ARtificial COLon (ARCOL) system and simulating the composition of human ileal effluents.

Components	Concentration (g/L)
Potato starch	5
Corn starch	2
Cellulose	1.5
Pectin	2
Arabinogalactan	1
Gum arabic	0.67
Guar gum	0.33
Inulin	1
Bacto peptone	2.5
Vegetable peptone	2.5
Bacto tryptone	5
Mucin type II	4
Yeast extract	4
Tween 80	1
Soy lecithin	0.375
Egg yolk	0.125
Bile salts	0.15
Bile extract	0.05
K_2_HPO_4_ 3H_2_O	1.14
NaCl	4.5
KCl	4.5
MgSO_4_ 7H_2_O	0.1
CaCl_2_ 2H_2_O	0.03
FeSO_4_ 7H_2_O	0.015
Hemin	0.005
L-cystein-HCl	0.3
NaHCO_3_	0.840
D-Pantothenic acid	1 × 10^−5^
Nicotinamide	5 × 10^−6^
4-aminobenzoic acid	5 × 10^−6^
Thiamin	4 × 10^−6^
Menadione	1 × 10^−6^
D-biotin	2 × 10^−6^
Vitamin K1	1 × 10^−6^
Vitamin B12	5 × 10^−7^
MnSO_4_ H_2_O	1.7 × 10^−4^
CoSO_4_ 7H_2_O	1.42 × 10^−4^
ZnSO_4_ 7H_2_O	1.44 × 10^−4^
CuSO_4_ 5H_2_O	2.5 × 10^−5^
NaWO_4_ 2H_2_O	3.3 × 10^−5^
H_3_BO_3_	6.2 × 10^−6^
Na_2_MoO_4_ 2H_2_O	2.4 × 10^−5^
NiCl_2_ 6H_2_O	2.4 × 10^−5^
Na_2_SeO_3_	3.8 × 10^−5^

**Table 2 microorganisms-09-00358-t002:** Selected criteria used to determine dysbiosis periods in ARtificial COLon (ARCOL) system.

Criteria	Cut-Off Levels for Differences	Control Condition
Gut microbiota activity		
Redox potential	≥ ± 200 mV	d-Day in treated bioreactor compared to d-Day in control experiment
NaOH consumption	Stop NaOH consumption	
Total gas production	Stop gas production	
CO_2_ concentrations	≥ ± 10%	
SCFA concentrations (acetate, propionate and butyrate)	≥ ± 25% for each SCFA	
Gut microbiota composition		
Total viable bacteria-FC	≥ ± 1 log	d-Day in treated bioreactor compared to d-Day in control experiment
Total bacteria-qPCR	≥ ± 1 log	
Richness	≥ ± 20%	d-Day compared to day 6 in treated bioreactor
Shannon	≥ ± 20%	
Bray-Curtis	≥0.5	

## Data Availability

All sequencing data were deposited to the National Center for Biotechnology Information Sequence Read Archive under accession number PRJNA642894.

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
