# Peer review of "An Oral FMT Capsule as Efficient as an Enema for Microbiota Reconstruction Following Disruption by Antibiotics, as Assessed in an In Vitro Human Gut Model"

_microorganisms, 2021, doi:10.3390/microorganisms9020358_

Round 1

Reviewer 1 Report

The manuscript described an innovative in vitro bioreactor method to evaluate the treatment effect of an oral FMT capsule on microbiota reconstruction following disruption by antibiotics. The study used a lot of experimental methods including molecular biology such as flow cytometry, qPCR, and Sr RNA gene sequencing to validate the results.  The manuscript was well writing with plenty of data. After answering the following questions, the manuscript is recommended for publication

  • Is there any previous studies, if any, on the correlation between this bioreactor model and in vivo models to show similar results to recover fecal-microbiota environment after antibiotic intervention or FMT treatment? If so, please include it in the discussion part. If not, please discuss it.
  • The design of the bioreactor in this study looks like fermentation of fecal samples, which could explained why capsule and enema (themselves are fecal samples) could recover richness and Bray-Curtis indexes at OTU level (Figure 5). However, does this fermentation process in bioreactor could simulate the fecal formation process in complex microbiota environment of colon in the physiological condition? The author should discuss it.
  • The title has some problem. In reviewer’s opinion, MINIBio Lab Reactor Bundles can not be named as in vitro human gut model because no human gut living cells are involved in this model.

Author Response

  • Is there any previous studies, if any, on the correlation between this bioreactor model and in vivo models to show similar results to recover fecal-microbiota environment after antibiotic intervention or FMT treatment? If so, please include it in the discussion part. If not, please discuss it.

All the parameters used to set-up our bioreactor model were established based on healthy human in vivo data thanks to an exhaustive literature review (the parameters are the following : pH, retention time, composition of ileal effluents entering the colon model). The model was further validated regarding gut microbiota composition and activity (SCFA and gaz production) compared to in vivo data in human, again under healthy conditions, i.e. without any disruptive elements added to the bioreactor. This was explained in the original version of this manuscript and completed in the revised paper (see lines 96 to 101, lines 526-528).

There is nonetheless no direct correlation between results obtained with our ARCOL model and in vivo data in animal or humans after antibiotic intervention or FMT treatment under similar experimental conditions (e.g. same subjects, same microbiota, same antibiotics and/or same FMT formulation). However, there are available studies using close operational (but not similar) conditions in the literature. For instance studies using capsules or enema exists but methods used (antibiotic used to create a dysbiosis, capsule load, antibiotic period of injection…) are different from the one we used. Therefore, we discuss the results obtained in the present study with publications from the literature using experimental conditions as close as possible from ours (see lines 544-547 and lines 602-607). To be clearer to Microorganisms readers we added both in the discussion on effects of antibiotics and FMT that our tested conditions are not fully similar to that described in the literature and discussed here (see lines 546-547 and 605).

  • The design of the bioreactor in this study looks like fermentation of fecal samples, which could explained why capsule and enema (themselves are fecal samples) could recover richness and Bray-Curtis indexes at OTU level (Figure 5). However, does this fermentation process in bioreactor could simulate the fecal formation process in complex microbiota environment of colon in the physiological condition? The author should discuss it.

Our model, as all available in vitro colon models, is inoculated with fecal samples because getting feces is an easy and non-invasive method to get microbiota (colon sampling is on the contrary an invasive procedure). Nonetheless, after inoculation we do not reproduce fecal fermentation but human colonic fermentation. Our in vitro colon model is set-up with the main physicochemical parameters specific from the colon (and not fecal) ecosystem, such as pH, retention time, nutrient availability, concentration and ratio of primary and secondary bile acids, anaerobiosis, redox values. This allows a shift not only from in vivo to in vitro situation, but also from fecal to colonic microbiota profiles.  All these information are available in section 2.2. To be clearer to Microorganisms reader, we added in this section several additional information that were missing (see lines 144-145) and clearly explain this shift in microbial profiles (see line 149-150). This shift in microbial profiles, including in the Firmicutes/Bacteroidetes ratio has been largely previously described  in other human colon models . A reference has been added to support our statement (Van Den Abbeele et al in 2010, Microbial community development in a dynamic gut model is reproducible, colon region specific, and selective for Bacteroidetes and Clostridium cluster IX).

  • The title has some problem. In reviewer’s opinion, MINIBio Lab Reactor Bundles can not be named as in vitrohuman gut model because no human gut living cells are involved in this model.

As explained in our answer to question 2, the term of in vitro human gut model is used because we reproduce the main microbial and physicochemical conditions found in a human colon, even if there is no living cells. This kind of appellation has been already used many times before for similar bioreactors developed by other groups in the world, also devoid of intestinal cells. It is for example the case of the SHIME model (previously mentioned with the article of Van Den Abbeele et al, 2010) or the PolyFermS model (described by Fehlbaum et al., 2019). We therefore think we can keep the terms used in our title in their current forms.

Reviewer 2 Report

In the current study, authors provide significant insight into the medically relevant dysbiosis in vitro human colon models as an alternative approach to in vivo assays during preclinical studies for evaluating FMT efficiency of the human gut ecosystem. The potential of this in vitro approach could be used in the management of many digestive or extra-intestinal pathologies during gut microbial dysbiosis. This study is relevant. The study design is straightforward, and it is a well written article with enough data to support their research thesis. I have few comments

Major Comments

In fecal sampling and FMT preparation of method section, how the equal microbiota addition was maintained by administration of fecal sample as 100 ml aliquots in each bioreactor?

Firmicutes (F) and bacteroid (B) are the dominant phyla and F/B ratio represents important index of health of gut microbiota. These studies  https://www.nature.com/articles/s41598-019-56561-1 , https://gutpathogens.biomedcentral.com/articles/10.1186/s13099-018-0230-4 and https://www.ncbi.nlm.nih.gov/pmc/articles/PMC4848671/  reported that a higher F/B ratio relates to pathological states such as obesity, metabolic syndrome as well as elderly age. In the current study, the gut microbiota profile of Donor 1/2 composed a F/B ratio of 70/30 or 65/35 which does not reflect a healthy gut. Explain?

The gut microbiota profile of Donor 1/2 composed a F/B ratio of 70/30 or 65/35, however the phyla profile of ARCOL showed run 1 and run 3 with 70-80% of bacteroid while run 2 with 40% of bacteroid. It may happen due to addition of ciprofloxacin. This need to be discussed in discussion part.

Is the SCFA profile is the mean from Run1, 2 and 3. If yes, how did the total SCFAs and gas production was affected by the F/B ratio from gut microbiota profile of Donor 1 and 2?

The difference in the anatomy of FMT infusion made FMT with colonoscopy (donor material administered mostly in the proximal colon or terminal ileum and/or cecum) superior to enema (donor material administered in the distal colon) leading to overall differences in cure rates. Moreover, although capsule FMT achieved normalized gut microbiome similar to colonoscopy, still its partially delayed due to multiple factors like the location of capsule release of FMT material, range in gastric pH, and intestinal transit time. https://link.springer.com/article/10.1007/s10620-020-06185-7#Sec17 . In the current study author should describe limitations of different modes of FMT administration e.g. capsule and enema

In line 563, ”In a preliminary study, we checked the  integrity of capsules in ileal effluents of the human TNO gastrointestinal TIM system [14]”. Author should briefly mention how integrity of capsules in ileal effluents was detected?

Researchers are focusing on, FMT administration from healthy donor to cancer patient where FMT is expected to enhance the effectiveness of immune checkpoint inhibitor (https://www.ncbi.nlm.nih.gov/pmc/articles/PMC7288675/ ) or FMT as a novel therapy for immune checkpoint inhibitor induced–colitis refractory to immunosuppressive therapy (https://ascopubs.org/doi/abs/10.1200/JCO.2020.38.15_suppl.3067 ). Author should discuss, how the proposed in vitro human colon model could reproduce the colon condition of cancer patients to detect FMT effectiveness during immune checkpoint inhibitor therapy.

Result section, line 310, “gas production restarted faster (1 or 2 days before ATB control) for all modes of administration, except for oral capsule in Run3. What could be the possible reason? 

Minor comments

Institutional review board (IRB) protocol number should be included in method section for collection of fecal samples from healthy adult human volunteers

The figure citations need to corrected in Figure 3 legend for 16S rRNA gene copies and viable bacteria count by flow cytometry in line 370 to 373.

The font size in line 70 and line 605-627 need to be changed to standard font of manuscript.

In the introduction section, author should include the limitations of other modes of FMT with references.

Information regarding use of germ-free mice to restore microbial diversity by FMT transplantation should be included.

In the antibiotic dosage  part of method section reference should be added to method used for detection of ciprofloxacin concentration in fermentative medium.

Figure 2. Effect of FMT treatment on gas and SCFA production. Mention y axis legend in sub-figures a to j

Author Response

Major Comments

  • In fecal sampling and FMT preparation of method section, how the equal microbiota addition was maintained by administration of fecal sample as 100 ml aliquots in each bioreactor?

As mentioned in the material and methods section (Line 115 to line 118), fecal samples were mixed with 500 mL of a 30 mM anaerobic sodium phosphate buffer (pH 6.5) supplemented with 1.9 mM cysteine and the suspension was filtered through a double layer of gauze. The fecal suspension was then divided into 100 mL aliquots that were rapidly transferred into each of the five bioreactors that were simultaneously inoculated. Mixing the fecal sample and dividing it just after allowed getting homogenous aliquots in each of the 5 bioreactor. This has been checked by comparing microbial profiles at initial times (by 16S Illumina sequencing) were all the profiles obtained in the five bioreactors were similar.

  • Firmicutes (F) and bacteroid (B) are the dominant phyla and F/B ratio represents important index of health of gut microbiota. These studies https://www.nature.com/articles/s41598-019-56561-1 , https://gutpathogens.biomedcentral.com/articles/10.1186/s13099-018-0230-4 and https://www.ncbi.nlm.nih.gov/pmc/articles/PMC4848671/  reported that a higher F/B ratio relates to pathological states such as obesity, metabolic syndrome as well as elderly age. In the current study, the gut microbiota profile of Donor 1/2 composed a F/B ratio of 70/30 or 65/35 which does not reflect a healthy gut. Explain?

We thank the reviewer for this fruitful comment. We agree with the referee comment on the Firmicutes/Bacteroidetes ratio but we can however underline that there is there is a lack of consensus in the literature on the relation between Firmicutes/Bacteroidetes ratio and healthy gut. This ratio can be widely variable between individuals and some publications clearly indicate that there is no direct link between this ratio and a pathology (see for instance the very recent publication from Magne et al in 2020 on obesity named The Firmicutes/Bacteroidetes Ratio: A Relevant Marker of Gut Dysbiosis in Obese Patients?)

Nonetheless, it is a well-known limitation of in vitro colon models where Bacteroidetes abundance is higher compared to in vivo data in humans. This was already observed in other colon models such as in the SHIME model (e.g. see Van Den Abbeele et al, 2010 Microbial community development in a dynamic gut model is reproducible, colon region specific, and selective for Bacteroidetes and Clostridium cluster IX). We know that this observation can represents a limit of in vitro colon model but in any case it can be associated with a pathological situation. To go further, we can inform the referee that some current optimizations are available in in vitro colon models (like in the SHIME or in our ARCOL model) to tackle this limitation (not available at the lab at the time of the experiments were performed). Such optimization implies the use of a mucosal phase inside the bioreactors where Firmicutes are dominant bacteria. With such optimization, considering the overall model (luminal and mucosal microbiota), we observed that the ratio Firmicutes/Bacteroidetes was improved compared to the in vivo situation in humans. We chose not to discuss this point in our paper because it was not the main purpose of the study and we feel it can lead the reader to lose the main objectives of the paper.

  • The gut microbiota profile of Donor 1/2 composed a F/B ratio of 70/30 or 65/35, however the phyla profile of ARCOL showed run 1 and run 3 with 70-80% of bacteroid while run 2 with 40% of bacteroid. It may happen due to addition of ciprofloxacin. This need to be discussed in discussion part.

We thank the reviewer for bringing this up to our attention. We do not exactly understand at which time of the fermentation the referee is talking about (T0, end of stabilization, after ATB treatment or after FMT treatment?). We will try to answer to the question as clearly as possible giving this limitation.

As a reminder, in this study, we used to make three biological replicates three fecal samples from 3 different donors. Run 1 corresponds to fermentations performed with the fecal sample from donor 1, Run 2 corresponds to donor 2 and Run 3 corresponds to donor 3. The composition of fecal microbiota is obviously donor dependent. For instance, at phylum level, corresponding to supplementary figure 1, we can see that the T0 of F1 are different according to the donor (e.g. Run). Donor 1 and 2 present a Firmicutes/Bacteroidetes ratio of approximately 65/35. On the opposite, donor 3 present a ratio Firmicutes/Bacteroidetes of 35/65. Therefore, only inter-individual variability on fecal inoculi can already explain part of the differences highlighted by the referee. This was already described in the manuscript lines 385-389).

When these fecal samples are introduced into the ARCOL, it is known that whatever the donor profile, as explained before, Bacteroidetes will take the advantage over Firmicutes, which can explain differences observed between T0 and later point collected during the fermentation. At the end of fermentation for the Control fermenter (no treatment was applied on this condition), ratios were approximately the same for the three donors (runs): 25-35/75-65 (see description lines 396-398). On the contrary, if we look at ATB control fermenters, we can see that at the end of the fermentation Donor 1 presents a ratio of 46/52, Donor 2 presents a ratio of 17/81 and Donor 3 presents a ratio of 25/74. Therefore, the impact of the ATB treatment in the bioreactor seems to be individual-dependent, as observed in vivo in humans (even if the microbiota was similarly stabilized in the bioreactors for the three donors). This can be an explanation for variations observed at phylum level at the end of ATB control fermentations of the different runs. As we did not clearly understand the question from the referee, up to now we did not add any additional discussion in the revised version.

  • Is the SCFA profile is the mean from Run1, 2 and 3. If yes, how did the total SCFAs and gas production was affected by the F/B ratio from gut microbiota profile of Donor 1 and 2?

Again, first we have to mention that it is not clear for us: at what time does the referee refer when he or she talks about F/B ratio (fecal samples, end of stabilization, after ATB treatment, after FMT treatment?) Given this limitation, we have done our best to answer to the referee question.

We first confirm to the referee that SCFA and gas profile in figure 2 is the mean from all three runs (donors). Looking at gas individual kinetics (not shown in the paper, but available on demand), we can see that the difference observed between donors in (fecal?) F/B ratio does not impact individual gas profiles for both Control and ATB control conditions. Concerning SCFA profile (not shown in the paper, available on demand), control condition presents similar values throughout the experiment for Donor 1 and Donor 3, approximately 60% of acetate, 25% of propionate and 15% of butyrate. Donor 2 presents lower values of acetate (45%) and higher values of butyrate (25%). Therefore it seems that difference in SCFAs profiles for the Control condition cannot be attributed to F/B ratio. On the contrary, there is no difference in SCFAs profiles for ATB control between donors, even if Donor 2 profile does not seem to stabilize after ATB end of treatment. Then, for ATB control condition also, we can establish no correlation between individual SCFA profiles and F/B ratio. Given our difficulty to fully understand the referee’s question and as we cannot establish relation between individual gas and SCFA profiles and individual F/B ratio, this point was not discussed in the revised version.

  • The difference in the anatomy of FMT infusion made FMT with colonoscopy (donor material administered mostly in the proximal colon or terminal ileum and/or cecum) superior to enema (donor material administered in the distal colon) leading to overall differences in cure rates. Moreover, although capsule FMT achieved normalized gut microbiome similar to colonoscopy, still its partially delayed due to multiple factors like the location of capsule release of FMT material, range in gastric pH, and intestinal transit time. https://link.springer.com/article/10.1007/s10620-020-06185-7#Sec17 In the current study author should describe limitations of different modes of FMT administration e.g. capsule and enema

We thank the referee for this useful comment. According to his/her suggestion, we added in the introduction section and in the discussion part some limitations associated with the different FMT routes. We therefore added the following sentences (from lines 64-67, 70-71, and 564-565): “These approaches are limited by difficulties to retain administered suspension (enema), risk of vomiting and aspiration as well as discomfort during administration (naso-gastric tubes), and necessary sedation and risk of perforation (jejunal infusion, colonoscopy)”. “Oral capsules offer the least invasive, cheapest and most easily stored administered form, and eliminate several procedural risks encountered with the other ways of FMT treatment, even if they are associated with risk of vomiting and aspiration, and if they can fail in reaching their target”. “Even if the approach can be limited by the difficulty for the patient to retain the suspension”. We also added a reference: Kyeong Ok Kim and Michael Gluck Fecal Microbiota Transplantation: An Update on Clinical Practice in 2019.

  • In line 563, ”In a preliminary study, we checked the  integrity of capsules in ileal effluents of the human TNO gastrointestinal TIM system [14]”. Author should briefly mention how integrity of capsules in ileal effluents was detected?

According to the referee suggestion, we detailed more in the revised version how the capsule integrity was checked in the TIM system. During those experiments, we first checked capsule integrity by visual control of the capsule (TIM is made of transparent glass compartments). We also performed bacterial numeration in the different intestinal compartments and in the ileal effluents to check if bacteria were released or not from the capsule. Those experiments confirm the integrity of the capsule up to the ileal compartment, which is in accordance with its design (caecum-release capsule). This was added in the revised version of the paper (see lines 576-577): ‘In a preliminary study, we checked the integrity of capsules in ileal effluents of the human TNO gastrointestinal TIM system by visual control and bacterial numeration’.

  • Researchers are focusing on, FMT administration from healthy donor to cancer patient where FMT is expected to enhance the effectiveness of immune checkpoint inhibitor (https://www.ncbi.nlm.nih.gov/pmc/articles/PMC7288675/ ) or FMT as a novel therapy for immune checkpoint inhibitor induced–colitis refractory to immunosuppressive therapy (https://ascopubs.org/doi/abs/10.1200/JCO.2020.38.15_suppl.3067 ). Author should discuss, how the proposed in vitrohuman colon model could reproduce the colon condition of cancer patients to detect FMT effectiveness during immune checkpoint inhibitor therapy.

We want to thank the reviewer for this perspective to our model. This could be indeed interesting to broaden our field of research to disease situation such as cancer. This was mentioned in the the paper from lines 634-635. We added in the revised version the references on cancer patients as suggested by the referee (Zhujiang Dai et al 2020; Yinghong Wang et al 2020). In addition to inoculation with faecal samples from patients, this would require a complete adaptation of all the physicochemical parameters of the in vitro colon models. This idea was added in the revised version of the paper (see lines 636-638): “to be fully relevant, a complete adaptation of the colon physicochemical parameters (pH, transit time, composition of nutritive medium and biliary salts) to the specific targeted population would be necessary.

  • Result section, line 310, “gas production restarted faster (1 or 2 days before ATB control) for all modes of administration, except for oral capsule in Run3. What could be the possible reason? 

We thank the referee for raising this interesting question. First, we can argue the fact that capsule contains lyophilized microbiota which can explain that it takes more time to restart gas production compared to enemas, as explained in the revised version of the paper lines 503-504: “This difference was mainly due to a high number of dysbiotic days for capsule in relation with acetate production (Figure 6) and probably results from latency time due to microbial revivification of capsule freeze-dried form”. This observation alone does not explain the difference mentioned by the reviewer because it concerns only capsule from Run 3 and not from Run 1 and Run 2. The second explanation is that, as explained earlier, each Run was performed following inoculation with a different donor fecal sample, so we cannot exclude a donor dependent effect. As the reasons remain speculative, we chose not to add in the discussion of the paper any explanation for this observation, except for latency time observed in microbial activity restart with the capsule form.

Minor comments

  • Institutional review board (IRB) protocol number should be included in method section for collection of fecal samples from healthy adult human volunteers

This study being a non-interventional study with no addition to usual clinical care, according to the French Health Public Law (CSP Art L 1121-1.1), the protocol does not require approval from an ethic committee.

  • The figure citations need to corrected in Figure 3 legend for 16S rRNA gene copies and viable bacteria count by flow cytometry in line 370 to 373.

We thank the referee for his/her comments and corrected the mistakes in the caption from figure 3 (see lines 378 to 382).

  • The font size in line 70 and line 605-627 need to be changed to standard font of manuscript.

According to referee observation, we changed the font size in the lines mentioned by the referee.

  • In the introduction section, author should include the limitations of other modes of FMT with references.

As the referee suggested, these limitations were added to the introduction of the manuscript (see answer to question 5).

  • Information regarding use of germ-free mice to restore microbial diversity by FMT transplantation should be included.

We thank the referee for this suggestion, we could not find references using germ-free mice to restore microbial diversity by FMT transplantation. Nonetheless, we already have introduced in the manuscript references concerning human microbiota-associated (HMA) mice since it is a frequently used strategy. These HMA mice are usually obtained by treating mice with a high dosage of antibiotics or by bowel cleansing.

  • In the antibiotic dosage part of method section reference should be added to method used for detection of ciprofloxacin concentration in fermentative medium.

According to the referee suggestion, we explain more in detail the method used to detect ciprofloxacin in the material and method section of our study. This method was adapted from the following articles, that have been added in the revised paper (lines 184-186): ‘Liquid chromatography-tandem mass spectrometry for the quantification of moxifloxacin, ciprofloxacin, daptomycin, caspofungin, and isavuconazole in human plasma; Julian Hösl, André Gessner, Nahed El-Najjar, 2018’; ‘A simple ultra-high-performance liquid chromatography-high resolution mass spectrometry assay for the simultaneous quantification of 15 antibiotics in plasma; S Lefeuvre, J Bois-Maublanc, L Hocqueloux, L Bret, T Francia, C Eleout-Da Violante, E M Billaud, F Barbier, L Got, 2017’.

  • Figure 2. Effect of FMT treatment on gas and SCFA production. Mention y axis legend in sub-figures a to j

According to referee suggestion we added y axis legend in subfigure a to j from Figure 2.

Round 2

Reviewer 2 Report

In this study, the authors describes a medically relevant dysbiosis in vitro human colon models as an alternative approach to in vivo assays during preclinical studies for evaluating FMT efficiency of the human gut ecosystem. This in vitro approach could be used in the management of many digestive or extra-intestinal pathologies during gut microbial dysbiosis. Overall, this is a well organized and  well written article with sufficient data to support relevance of this research.

Author Response

We thank the reviewer for giving us the opportunity to improve the english of our manuscript. We then carefully revised the manuscript for english spelling.